

# Common motor patterns of asymmetrical and symmetrical bipedal gaits

Germán Pequera[1,2], Ignacio Ramírez Paulino[3] and Carlo M. Biancardi[2]

[1] Ingeniería Biológica, CENUR Litoral Norte, Universidad de la República, Paysandú, Uruguay
[2] Biomechanics Lab., Dept. de Ciencias Biológicas, CENUR Litoral Norte, Universidad de la República, Paysandú, Uruguay
[3] Inst. de Ingeniería Eléctrica, Fac. de Ingeniería, Universidad de la República, Montevideo, Uruguay

## ABSTRACT

**Background**. Synergy modules have been used to describe activation of lower limb muscles during locomotion and hence to understand how the system controls movement. Walking and running have been shown shared synergy patterns suggesting common motor control of both symmetrical gaits. Unilateral skipping, an equivalent gait to the quadrupedal gallop in humans, has been defined as the third locomotion paradigm but the use by humans is limited due to its high metabolic cost. Synergies in skipping have been little investigated. In particular, to the best of our knowledge, the joint study of both trailing and leading limbs has never been addressed before.

**Research question**. How are organized muscle activation patterns in unilateral skipping? Are they organized in the same way that in symmetrical gaits? If yes, which are the muscle activation patterns in skipping that make it a different gait to walking or running? In the present research, we investigate if there are shared control strategies for all gaits in locomotion. Addressing these questions in terms of muscle synergies could suggest possible determinants of the scarce use of unilateral skipping in humans.

**Methods**. Electromyographic data of fourteen bilateral muscles were collected from volunteers while performing walking, running and unilateral skipping on a treadmill. Also, spatiotemporal gait parameters were computed from 3D kinematics. The modular composition and activation timing extracted by non-negative matrix factorization were analyzed to detect similarities and differences among symmetrical gaits and unilateral skipping.

**Results**. Synergy modules showed high similarity throughout the different gaits and between trailing and leading limbs during unilateral skipping. The synergy associated with the propulsion force operated by calf muscles was anticipated in bouncing gaits. Temporal features of synergies in the leading leg were very similar to those observed for running. The different role of trailing and leading legs in unilateral skipping was reflected by the different timing in two modules. Activation for weight acceptance was anticipated and extended in the trailing leg, preparing the body for landing impact after the flight phase. A different behaviour was detected in the leading leg, which only deals with a pendular weight transference.

**Significance**. The evidence gathered in this work supports the hypothesis of shared modules among symmetrical and asymmetrical gaits, suggesting a common motor control despite of the infrequent use of unilateral skipping in humans. Unilateral skipping results from phase-shifted activation of similar muscular groups used in symmetrical gaits, without the need for new muscular groups. The high and anticipated

Corresponding author
Germán Pequera,
gpequera@cup.edu.uy

muscle activation in the trailing leg for landing could be the key distinctive event of unilateral skipping.

## INTRODUCTION

Although walking and running are the most used gaits, the repertoire of bipedal locomotion also includes skipping, a bipedal gait equivalent to gallop (*Saibene & Minetti, 2003*). Unlike walking and running, skipping is an asymmetric gait, where one foot behaves differently from the other. We can distinguish a trailing limb (the first of a pair to strike the ground) from a leading limb (*Hildebrand, 1977*). Also, we identify two unilateral skipping forms, equivalents to the quadrupedal right- and left-leading transverse gallop, and a bilateral form, equivalent to the quadrupedal rotary gallop (*Minetti, 1998*; *Biancardi & Minetti, 2012*).

The mechanical model of skipping-gallop is a double pogo-stick, with a step vaulting phase (similar to walking) followed by a bouncing phase (as occurs in running). These features are distinctive enough for some authors to refer to skipping as the "third paradigm of locomotion" (*Minetti, 1998*; *Saibene & Minetti, 2003*). Skipping is energetically demanding with respect to its quadrupedal homologous (*Minetti et al., 1999*; *Biancardi & Minetti, 2012*), or compared to walking and running (*Minetti, 1998*; *Ackermann & Van denBogert, 2010*; *Minetti, Pavei & Biancardi, 2012*; *Pavei, Biancardi & Minetti, 2015*), and therefore seldom used by humans. This studies commonly have been addressed metabolic and mechanical parameters in different conditions of speed and gravity to explain the high metabolic power developed by humans during this gait. However, very few studies have performed muscle activation analysis to investigate how similar are the activation patterns outputs between this gait and common symmetrical gaits (*Ivanenko et al. 2008*).

Muscle activation patterns during locomotion have been described by means of muscle synergies (or motor modules) (*Cappellini et al., 2006*; *Ting & McKay, 2007*). Each synergy represents the activity of the co-activated muscles triggered by a single control signal. This scenario suggests the existence of a control mode in a low-dimensional space, rather than a specific control command for each muscle. Although the role and existence of such modules are still questioned (*Tresch & Jarc, 2009*), the muscle synergy hypothesis provides a good interpretation of the neural control strategy for activating muscles during locomotion (*Lacquaniti, Ivanenko & Zago, 2012*; *Ting et al., 2015*; *Yokoyama et al., 2019*). Synergy identification has been done based on electromyography signals (EMG) from muscles of the limbs; these groups of signals were then decomposed by applying various computational methods (*Tresch, Cheung & d'Avella, 2006*).

*Cappellini et al. (2006)* showed that, while walking or running on a straight path at a different controlled speed, the lower limbs muscles group in similar five modules; the only difference between walking and running was the temporal activation of one of the

modules. Being walking and running symmetric, the authors assumed that the synergies would have been the same in the two limbs, and thus measured the electromyographic signals on the muscles of only one limb (*Cappellini et al., 2006*). The same approach was followed by other authors, which confirmed the shared basic synergy modules at different inclinations in walking (*Saito et al., 2018a*), and in running (*Saito et al., 2018b*); the same was observed in walking *vs.* cycling (*Barroso et al., 2014*). Different authors associated each synergy module to a particular step phase (*Cappellini et al., 2006*; *Neptune, Clark & Kautz, 2009*; *Santuz et al., 2018b*). However, to the best of our knowledge, no synergy studies have been conducted considering both the leading and trailing limbs in asymmetric gaits like unilateral skipping. In this sense, studying the unilateral skipping asymmetries in terms of muscular synergies could help to identify common underlying control mechanisms, shared among gaits.

Our main hypothesis was that the synergy modules should be shared by both symmetric and asymmetric gaits, with even timing differences between trailing and leading limbs of unilateral skipping. Based on this, the following working hypotheses have been defined:

- Walking, running and unilateral skipping presents shared muscle synergies, with differences in their activation timing;
- The activation timings of the trailing and the leading limbs of skipping differ;
- According to the mechanical model of skipping as a combination of pendulum and spring-mass, the activation timings of the trailing and leading legs may be more similar to walking or running; alternatively, the activation of one limb may be similar to that of walking while the other may be similar to that of running.

## MATERIALS & METHODS

### Participants
Fourteen healthy men took part in the experiment (age: 25.3 ± 3.7 years, weight: 78.15 ± 13.8 kg, height: 1.76 ± 0.09 m). We excluded subjects with muscular pain, cardiovascular or neuromuscular diseases.

All the participants signed a written informed consent according to the Declaration of Helsinki. The project was approved by the Ethics Committee of the Centro Universitario Regional Litoral Norte, Universidad de la República (Exp. No 311170-000521-18).

### Experimental protocol
The protocol was completed in a single session. At the beginning, subjects warmed up and practiced unilateral skipping overground and on the treadmill until they were familiar with the task. Afterwards, they were asked to walk, run and skip unilaterally at controlled speeds on a treadmill. Ten trials of four minutes were conducted: walking at 0.83, 1.11, 1.39, 1.81 m s$^{-1}$; running at 1.81, 2.50 and 3.06 m s$^{-1}$; and unilateral skipping at 1.39, 1.81 and 2.50 m s$^{-1}$. These speeds were chosen based on previous experiences (*Minetti, Pavei & Biancardi, 2012*; *Pavei, Biancardi & Minetti, 2015*), with some overlap so that different gaits could be compared at the same velocities.

The order of the speeds were randomized within each gait condition. In order to avoid muscle fatigue, there was a three minute rest between trials.

## Data acquisition

Electromyographic signals were bilaterally recorded from *Gluteus Medius* (*GluM*), *Bíceps Femoris* (long head, *BF*), *Rectus Femoris* (*RF*), *Vastus Medialis* (*VM*), *Tibialis Anterior* (*Tib*), *Gastrocnemius Medialis* (*GM*) and *Soleus* (*Sol*) during the last 90 s of each trial. The electrode placement in each muscle was realized following the SENIAM protocols (*Stegeman & Hermens, 2007*). This muscle set was chosen based on the analysis of muscle synergies locomotion studies when less than ten muscles by leg were measured (*Clark et al., 2010*; *Barroso et al., 2014*; *Pérez-Nombela et al., 2017*; *Shuman et al., 2019*). Signals were sampled at 2 kHz by a Delsys Trigno EMG system (Boston, MA, USA), bandpass filtered between 20 Hz to 400 Hz, and amplified (x10000). Each Trigno unit includes a single differential EMG sensor synchronized to a triaxial accelerometer.

In the same time window, kinematic data were recorded by an eight-camera MOCAP system (Vicon, Oxford Metrics), at 100 Hz. Eighteen markers were located on the main joint centers in order to reconstruct the 3D movement with a reliable biomechanics model composed of eleven segments (*Pavei et al., 2017*).

## Kinematic analysis

Accelerometric signals from the *Tib* sensor were used to identify heel strikes (*Oliveira et al., 2016*). Stride frequency (*Fs*) was computed as the reciprocal of the cycle duration, while the stride length (*Ls*) was obtained by dividing *Fs* into the progression speed. The duty factor (*DF*) (*Alexander, 2003*) that is, the fraction of the total cycle in which a given foot is on the ground, was computed by identifying the times of landing and take-off of the heel and fifth metacarpal markers (*Pavei, Biancardi & Minetti, 2015*). Flight times ($F_T$) during skipping and running were calculated using the same markers.

The body Center Of Mass (*COM*) positions were determined as the weighted mean of the centers of mass of the segments, which were in turn determined using anthropometric Dempster tables (*Winter, 2003*). The trajectory of the *COM* 3D position within a stride, in local coordinates (as displacing on a treadmill), would describe a closed curve (*Minetti, 2009*; *Minetti, Cisotti & Mian, 2011*). In symmetrical gaits, the trajectory exhibits a double loop, each one representing one step, including the stance phase of one limb and the swing phase of the other (*Minetti, Cisotti & Mian, 2011*).

## EMG processing

The dominant leg of each subject was assessed before the experimental session (*Van Melick et al., 2017*). The EMG of the dominant limb during walking and running, and those of both limbs during unilateral skipping were processed. By spontaneous choice of each participant, the dominant leg was used by all participants as trailing leg during unilateral skipping. EMG were full-wave rectified and low-pass filtered (Butterworth, 4th order, cutoff frequency = 10 Hz). In each trial, 20 strides were normalized in amplitude by the peak activation signal of the gait cycle, resampled at 1% of the gait cycle and concatenated, resulting in a matrix $EMG_o$ ($m \times t$), where $m = 7$ (the number of muscles) and $t = 2000$ (number of samples). A non-negative matrix factorization (*NNMF*) algorithm was applied to the matrix in order to identify the muscle synergies (*Lee & Seung, 1999*; *Ting &*

*Macpherson, 2005*). Under this model, the $EMG_o$ matrix can be decomposed (factorized) as:

$$EMG_o = W \times C + R = EMG_r + R$$

The number of synergies identified by the model is given by the number of columns of the matrix $W$ ($m$ x $n$), which coincides with the number of rows in the matrix $C$ ($n \times t$). The i-th synergy is described by the i-th column of $W$, which specifies the relative weight of the $m$ muscles within the synergy, and the corresponding i-th row of $C$, which describes the activation pattern of the synergy across time. $EMG_r$ is the reconstructed EMG matrix while the matrix $R$ ($m \times t$) is considered a residual error of the model.

To reduce the effect of local minima, the *NNMF* algorithm was executed 40 times with random initial conditions and the factorization with the lowest squared reconstruction error was selected.

To identify the number of synergies, *NNMF* was applied on bootstrapped EMG signal considering a different number of synergies each time, from 1 to 7. $EMG_o$ of five randomly chosen subjects in each condition was concatenated ($EMG_b$). This procedure was repeated 20 times in each trial to estimate confidence intervals of the variance accounted for (VAF) for each number of synergy. The VAF defined as:

$$VAF = 1 - \frac{\|EMG_b - EMG_{rb}\|_{F^2}}{\|EMG_b\|_{F^2}}$$

where $F$ represents the Frobenious norm and $EMG_{rb}$ represents $EMG_b$ reconstruction from *NNMF*. VAF *vs* synergy number plots were used to determine the appropriate number of synergies for each condition. The point at which the VAF curve shows a considerable slope change is determined as the appropriate synergy number. As *Cheung et al. (2005)* report, the slope of the VAF plot usually decreases gradually with the synergy number, creating difficulties to estimate, by visual inspection, the number of synergies correctly. For hence, the number of synergies was identified from a linear regression method already used by several authors (*Cheung et al., 2005*; *d'Avella et al., 2006*; *Santuz et al., 2018b*). This method identifies the smallest number of synergies such that a linear fit of the VAF curve, from 1 to 7, had a residual mean square error of less than $1 \times 10^{-4}$.

The weightings were sorted using the maximal cosine similarity (*cosim*) (*Hagio & Kouzaki, 2014*; *Hagio, Fukuda & Kouzaki, 2015*; *Banks et al., 2017*). Given two weight vectors $a$ and $b$, their *cosim* is given by

$$cosim(a, b) = cos\theta = \frac{\sum a_i b_i}{\|a\| \|b\|}$$

that is the cosine of the angle between the two weight vectors.

*cosim* was calculated between weightings modules of an arbitrary subject and the rest of subjects employing an iterative process. We defined the pair of modules with the highest *cosim* value as the same group. (*Hagio & Kouzaki, 2014*; *Torres-Oviedo & Ting, 2007*). *cosim* was also used to measure the similarity between motor modules across different gaits; values of *cosim* above 0.6 commonly indicate similarity between motor modules (*d'Avella, Saltiel & Bizzi, 2003*).

The temporal features of module activation were evaluated using a center of activation (*CoA*). *CoA* represents the center of mass of the circular distribution of the temporal components in polar coordinates ($0 \leq \theta_j \leq 2\pi$) (*Santuz et al., 2018a*; *MacLellan, 2017*), and is given by $CoA = tan^{-1}\left(\frac{B}{A}\right)$ where

$A = \sum_{j=1}^{100}\left(cos\theta_j \times C_j\right)$ and $B = \sum_{j=1}^{100}\left(sin\theta_j \times C_j\right)$

where *Cj* is the amplitude of the activation pattern in sample *j*.

## Statistical analysis

Differences in spatiotemporal parameters and in the number of synergies between gaits were evaluated using a one-way ANOVA, or Kruskal–Wallis test if Shapiro–Wilk tests on the residuals report non-normality. T-tests (or Wilcoxon rank-sum test, in case of non-normality) with Bonferroni correction were applied to reduce the risk of Type I errors in multiple hypothesis testing scenarios. One sample *t*-test was realized on *cosim* values in each condition comparison to check if synergy pairs were significantly larger than the 0.6 (similarity threshold). The *CoA* of different gaits for fixed speeds were compared using the Watson-Williams test for circular data. Statistical significance was assessed at 0.05.

## RESULTS

The displayed results show only comparisons at 1.81 m s$^{-1}$, where the three gaits are present. Comparisons at 1.39 m s$^{-1}$ and at 2.5 m s$^{-1}$ can be found in supplementary material.

### Gait parameters

Table 1 resumes the spatio-temporal parameters of all the analysed gaits. *Ls* and *Fs* presents statistical differences between all gaits (all *p*-values equal or less than 0.001).

The *DF* of the trailing and leading limbs of unilateral skipping was significantly different at 1.81 m s$^{-1}$. *DF* during unilateral skipping in the leading leg did not displayed statistical differences with running (posthocs *p*-values near 1 at 1.81 m s$^{-1}$). *DF* in walking showed differences with all gaits and legs. Significant *p*-values less than 0.01 were reported between walking and both unilateral skipping legs, when multiple comparisons were realized at 1.81 m s$^{-1}$. Table 1 reveals also the unilateral skipping and running flight times as fraction of stride times. *FT* in unilateral skipping were on average longer than in running showing statistical significance ($p < 0.001$) at 1.81 m s$^{-1}$. Longer *FT* imply greater average vertical ground reaction force during the contact phase of unilateral skipping, as pointed out by *Minetti, Pavei & Biancardi (2012)*.

### EMG envelopes

During walking, muscles of Triceps Surae were active at mid-stance, *RF* and *Vas* were active finalizing swing phase and at the beginning of stance phase, *Tib* was active mostly during the swing; while *BF* and *GluM* show activation during the late swing and at early stance (Fig. 1). In running, the extensor muscles present EMG peak activation at the early stance phase while *Tib*, *BF* and *GluM* presents similar activation patterns than walking.

During skipping we observed different muscle activation patterns between both legs, mostly in *Tib*, *RF* and *GluM* muscles (Fig. 1). *Tib* shows peak activation at the early swing

Pequera et al. (2021), *PeerJ*, DOI 10.7717/peerj.1970

**Table 1 Spatiotemporal patterns of walking, running and unilateral skipping at 1.81 m s$^{-1}$.** In case of parametric test, ANOVA and t-test with Bonferroni correction tests were used. In case of non-parametric, Kruskal-Wallis and Wilcoxon rank-sum test (also with Bonferroni correction) was used. Significant differences between gaits were established at pvalues <0.05.

| | ANOVA (or Kruskal–Wallis) | | | Gait (or leg) | | | | Post-hoc (Gait A vs Gait B) | | |
|---|---|---|---|---|---|---|---|---|---|---|
| | Parametric | F (or H) | p | W | R | S$_{trailing}$ | S$_{leading}$ | Gait A | Gait B | p |
| Duty Factor | False | 35.98 | <0.01 | 0.58 ± 0.02 | 0.45 ± 0.07 | 0.28 ± 0.06 | 0.44 ± 0.04 | R | S$_{trailing}$ | 0.002 |
| | | | | | | | | R | S$_{leading}$ | 1 |
| | | | | | | | | R | W | 0.002 |
| | | | | | | | | S$_{trailing}$ | S$_{leading}$ | 0.001 |
| | | | | | | | | S$_{trailing}$ | W | <0.001 |
| | | | | | | | | S$_{leading}$ | W | <0.001 |
| Stride Frequency (Hz) | False | 32.19 | <0.01 | 1.13 ± 0.07 | 1.32 ± 0.08 | | 1.54 ± 0.2 | R | W | <0.001 |
| | | | | | | | | S | W | <0.001 |
| | | | | | | | | R | S | 0.001 |
| Stride Length (m) | True | 53.79 | <0.01 | 1.6 ± 0.09 | 1.38 ± 0.08 | | 1.19 ± 0.13 | R | S | 0.001 |
| | | | | | | | | R | W | <0.001 |
| | | | | | | | | S | W | <0.001 |
| Flight Time (fraction of cycle) | True | – | – | – | 0.17 ± 0.07 | | 0.31 ± 0.07 | S | R | <0.001 |
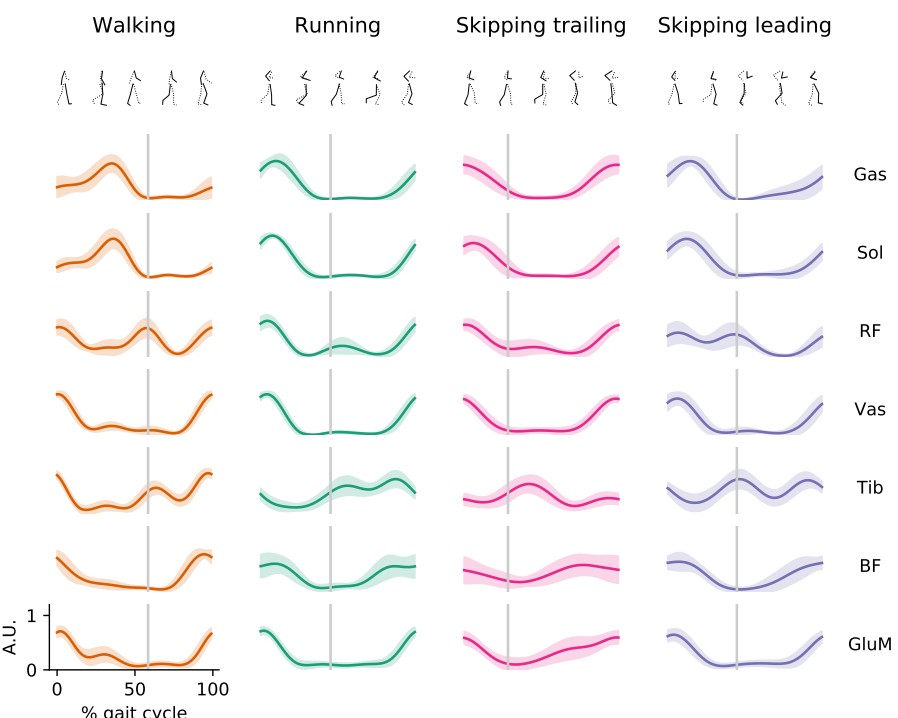

**Figure 1** **EMG envelopes.** Averaged ($n = 14$) electromyographic linear envelopes of 7 recorded muscles for each gait as a function of the gait cycle at 1.81 m s$^{-1}$. For each subject, linear envelope from each muscle were previously normalized by the peak throughout 20 cycles. Muscle abbreviations: *Gastrocnemius Medialis (Gas), Soleus (Sol), Rectus Femoris (RF), Vastus Medialis (Vas), Tibialis anterior (Tib), Biceps Femoris (BF) and Gluteus Medius (GluM)*. A.U., arbitrary unit.

in the trailing leg, while in the leading leg this muscle seems to activate at toe-off and late swing. In *RF* can be

seen a double peak in the EMG envelope of the leading leg; one at early stance and other at the toe-off. In trailing leg *RF* is not activated at toe-off. *GluM* present activation at the early stance and a slow but sustained increment of activation during swing phase in trailing leg, distinct to leading leg where *GluM* present a similar pattern to running with activation at the early stance and late swing. The rest of the muscles present similar activation patterns shape but with anticipated activation when trailing leg is compared with leading leg in skipping.

## Muscle synergies

Figure 2 shows the number of synergies required in each gait at 1.81 m s$^{-1}$. When the number of synergies were evaluated in different gaits, Kruskal-Wallis test did not identify differences ($p = 0.068$). Around 3 synergies, in average, were necessary to meet the reconstruction quality criteria used in this work in all conditions (Fig. 2).

As Fig. 2 points out, the mean of the number of synergies was greater than 3 in some conditions. In order to compare all gaits and speeds, 4 synergies were used in all subsequent analyses (*Clark et al., 2010*; *Barroso et al., 2014*; *Pérez-Nombela et al., 2017*; *Shuman et al.,*

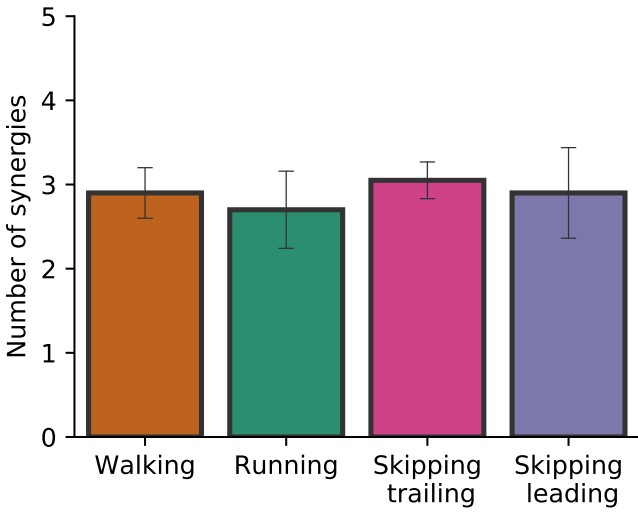

**Figure 2** **Number of synergies.** Number of synergies (mean ± s.d.) between gaits at 1.81 m s$^{-1}$. No significant differences were found ($p = 0.07$).

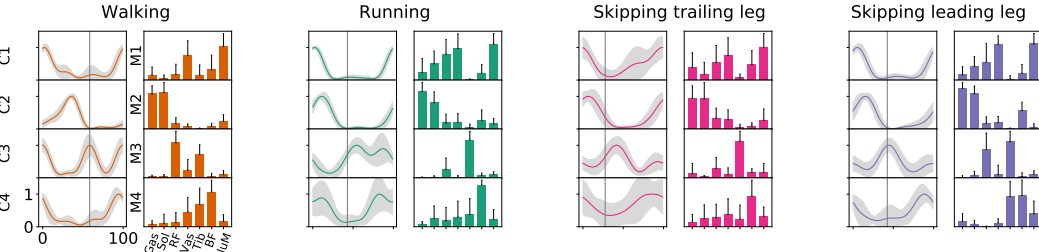

**Figure 3** **Activation timings profiles and muscle synergies of walking, running and skipping (both legs) at 1.81 m s$^{-1}$.** Solid lines in activation timing profiles indicate average ($n = 14$) and shaded area represent ± s.d. Vertical lines divide the stance (on left) and swing (on right) phases. Muscle weightings are represented by bars plot. Muscle abbreviations: *Gastrocnemius medialis (Gas), Soleus (Sol), Rectus Femoris (RF), Vastus Medialis (Vas), Tibialis anterior (Tib), Biceps Femoris (BF)* and *Gluteus Medius (GluM)*.

*2019*). The modules were composed basically by the same muscles in all gaits. Module 1 included *GluM*, *RF* and *VL*; module 2 included Triceps Surae muscles; module 3 covered Tibial muscle; module 4 comprised *BF* (Fig. 3). When muscle weightings were compared across gaits, similarity was observed across all gaits and legs (Fig. 4): in most comparisons, 3 of the 4 modules presented highly similar mean values in all conditions; however, synergy 4 appears to be specific to each locomotor pattern. On the other hand, the trailing leg in unilateral skipping displayed only one shared synergy when compared to walking at 1.81 m s$^{-1}$.

## Synergy temporal patterns

Figure 5 shows the mean *CoA* of each module across gaits at 1.81 m s$^{-1}$. Some significant differences can be observed. Synergy 1 for unilateral skipping trailing leg shows late *CoA*
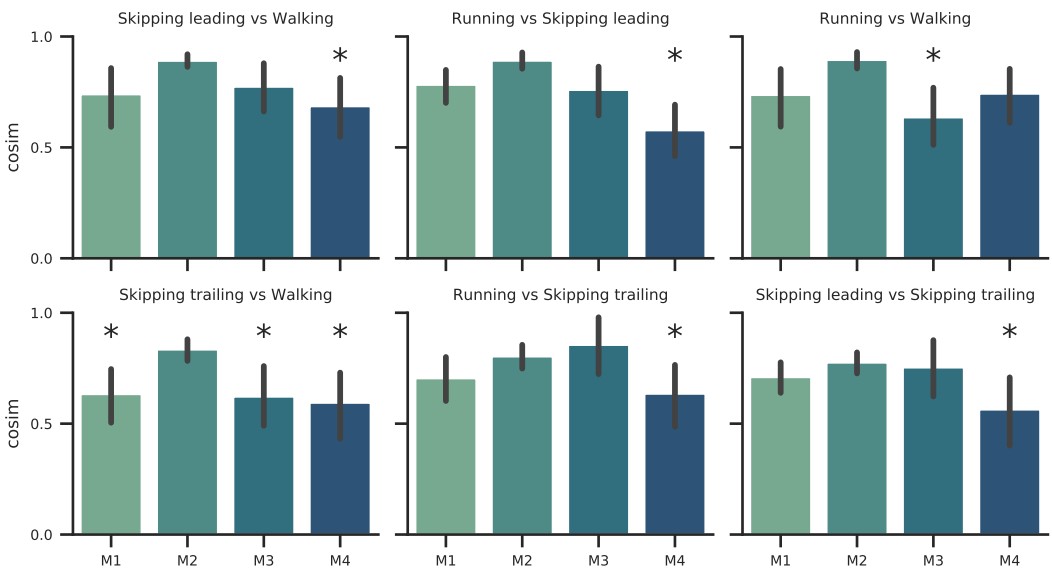

**Figure 4   Synergy similarities between gaits at 1.81 m s⁻¹.** Cosine similarities (*cosim*) means (*n* = 14) ± s.d across gaits comparisons. * indicate synergies that do not reach the similarity threshold.

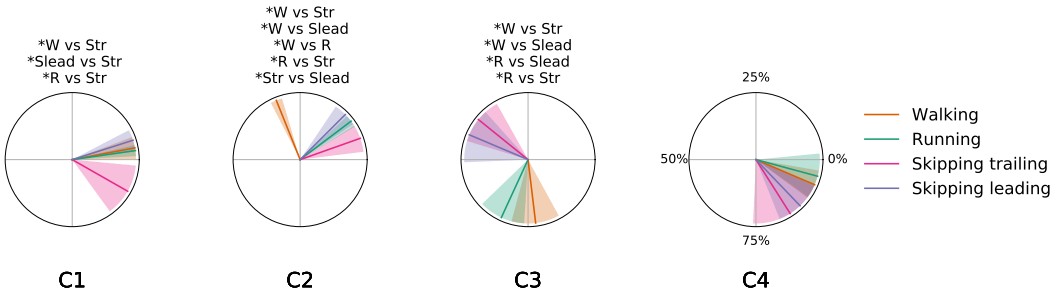

**Figure 5   Centers of activation in different gaits at 1.81 m s⁻¹.** Averaged (*n* = 14) centers of activation *(CoA)* ± s. d. of each motor module plotted in polar coordinates. Significant differences between gaits are indicated above each plot. Abbreviations: Running (R), trailing leg in skipping (Str), leading leg in skipping (Slead) and walking (W).

and significant differences with all gaits (Fig. 5). *CoA* in walking, running and leading leg in unilateral skipping did not present statistical differences between them for this synergy. For the propulsion synergy (module 2), we can identify 3 groups of *CoA* classified by statistical differences (Fig. 5). One group includes an anticipated activation of synergy 2 in trailing leg, another group includes running and leading leg in skipping *CoA* and the third group shows a delayed *CoA* in synergy 2 for walking (Fig. 5). Synergy 3 shape two groups based in statistical test application. One anticipated group of *CoA* s composed by both legs of unilateral skipping and other group composed by symmetrical gaits showing a belated *CoA* (Fig. 5). Synergy 4 did not present statistical differences between gaits (Fig. 5).

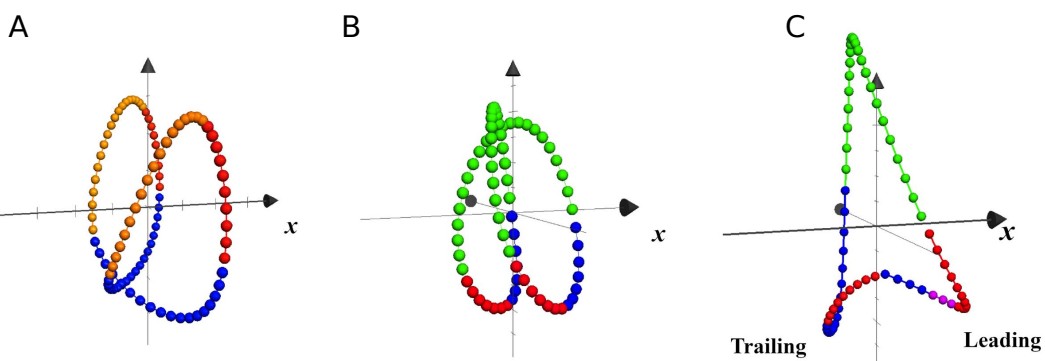

**Figure 6 Trajectories of the center of mass in local coordinates of one representative subject at 1.81 m s$^{-1}$.** Each closed loop corresponds to one stride of walking (A), running (B) and skipping (C), at different scale. The $x$ axis indicates progression direction. Blue: weight acceptance (synergy module 1); Red: propulsion (synergy module 2); Violet: modules 1 & 2 in leading skipping; Yellow: pendular transference in walking; Green: ballistic phase in running and skipping.

## Center of mass trajectory

In Fig. 6 are plotted the trajectories of the *COM* during three single strides, taken as examples of walking, running and unilateral skipping. According to the kinematics of the *COM* movement, the lower parts of the loops represent the phases of heel strike and acceptance of the weight for the stance limb, followed by the propulsion phase at the moment that the *COM* is rising. The colors highlight the effects of the activation of the first two synergy modules on the stance limb.

In unilateral skipping, the lower vertices of the triangular loop represent, as indicated in the figure, the trailing and leading stance phases. A clear asymmetry can be seen of the synergistic activation during unilateral skipping (Fig. 6).

## DISCUSSION

### Gait parameters

During unilateral skipping, the trailing and leading legs play different roles, the former being more dedicated to vertical braking and forward propulsion, and the latter vice versa, to forward braking and vertical propulsion (*Fiers et al., 2013*). Considering the high average vertical force observed during the stance phase of unilateral skipping, in accordance with *Minetti, Pavei & Biancardi (2012)*, and the ability of the muscles to absorb negative work, it is reasonable to expect that the *DF* of the trailing limb will be lower than that of the leading limb, as detected by *Fiers et al. (2013)*. Our results confirm this trend (Table 1).

The *DF* of the leading leg was not significantly different from that of running at the same speed. The $F_S$ of running was on average 1.32 Hz, near the optimal value of 1.4 Hz (*Snyder & Farley, 2011*). With respect to running, unilateral skipping displayed significantly higher $F_S$ and significantly lower $L_S$ at the same speed (Table 1). This confirms the previously observed pattern (*Minetti, Pavei & Biancardi, 2012*; *Pavei, Biancardi & Minetti, 2015*; *Fiers et al., 2013*). Higher $F_S$ can lead to higher energy expenditure, which is one of the characteristics of unilateral skipping with respect to running (*Snyder & Farley,*

*2011*; *Pavei, Biancardi & Minetti, 2015*). One reason for that could be related to the necessity to increase the muscle shortening velocities, in order to achieve higher $F_S$ (*Lindstedt et al., 1985*).

## Muscle synergies

To the best of our knowledge, this was the first simultaneous evaluation of synergies of the trailing and leading limbs in unilateral skipping. Different works on muscle synergies during locomotion reported 4–5 synergies to reconstruct the original EMG signals (*Cappellini et al., 2006*; *Yokoyama et al., 2016*; *Santuz et al., 2018a*), while our work shows an average of around 3 in all conditions (Fig. 3). This result could be explained by two reasons. First, we recorded a smaller set of muscles (7 by leg) than other works (*Cappellini et al., 2006*; *Yokoyama et al., 2016*; *Santuz et al., 2018a*). Some authors demonstrated that the number of muscles measured affects the VAF values, overestimating them when the number of EMG signals diminish (*Steele, Tresch & Perreault, 2013*). In fact, Pérez-Nombela and collaborators (2017) obtained a number of synergies comparable to our work, measuring seven muscles by side in the control group. The other reason is that there is no agreement about a standard method to estimate the number of synergies from the VAF curve (*Turpin, Uriac & Dalleau, 2021*), and this fact creates difficulties to compare results associated with this parameter.

According to *Santuz et al. (2018a)* and *Cappellini et al. (2006)*, the same number of synergies can describe walking and running. Our results confirm those conclusions, adding unilateral skipping (both trailing and leading limbs) to the gait cluster (Fig. 3). The number of synergies has been interpreted as a locomotor output complexity indicator, where a greater number of modules reflects more complexity (*Clark et al., 2010*). Based on these results we suggest that unilateral skipping does not imply differences in motor complexity when compared with common symmetrical gaits. For hence, the disuse of unilateral skipping would be not determined by different complexity in the task execution.

According to our first hypothesis, we found the number and the composition of the motor modules to be similar across the repertoire of human locomotion, including both legs during the asymmetrical unilateral skipping gait as Figs. 3 and 4 shows. Synergy modules from 1 to 3 were shared in all gaits and legs, while the fourth module revealed lower values of similarity Fig. 4. Yet, the *BF* was always the principal component of module 4 (Fig. 4).

In unilateral skipping, both legs displayed weighting coefficients similar to each other and to the symmetrical gaits motor modules (Fig. 4). These similarities across the different locomotor patterns suggest a common control strategy for symmetric and asymmetric gaits in humans. The above idea was originally proposed by *Whitall & Caldwell (1992)* when they observed similar intralimb coordination, however under different interlimb coordination, where unilateral skipping was compared to running. So, an important finding of this work is the evidence of an shared control scheme between all human gaits. Similarities between the module weighting coefficients of walking and running was already observed in previous investigations (*Cappellini et al., 2006*); their results underlined higher correlations when the motor modules of the lower limbs muscles were compared in walking and running.

Conversely, lower correlations appeared when including the upper limbs and trunk muscles in the analysis. Some authors detected differences among the synergies of different gaits and speeds, attributing this difference to the methodology used (*Yokoyama et al., 2016*). Recently, a walking and running neuromusculoskeletal model has been developed based on parameters of the muscle synergy hypothesis, confirming that both gaits can be produced by the same set of modules with different activation timing (*Aoi et al., 2019*). According to our results, it should be possible to make this model work for unilateral skipping as well.

Unilateral skipping is homologous to the transverse gallop of quadrupeds, which is their most economical gait at high speeds. The fact that synergies were shared in all three gaits, including unilateral skipping, highlights the idea of the synergies like phylogenetically conserved structures, discussed by other authors (*Dewolf et al., 2020*). However, despite preserved neural circuitry in evolution of locomotion, the efficiency of gait patterns varies in function of the different biomechanical and anatomical characteristics of the species, and can be a strong selective factor for or against the employment of gait patterns.

## Synergy temporal patterns

As a summary we can say that walking and running are only differentiated by an anticipated activation of the Triceps Surae muscles; leading leg during unilateral skipping only is differentiated of running by anticipated activation of synergy 3 and *CoA* in synergy 3 is able to distibguish symmetrical and asymmetrical gaits. The temporal pattern of trailing leg during unilateral skipping appears very different to those of symmetrical gaits (3 out of 4 *CoA* were different in the majority of the comparisons). The above results indicate that the movements of the trailing leg in unilateral skipping was not similar to those of walking or running. On the other hand, our comparison of trailing and leading legs in unilateral skipping yielded 2 different *CoA* across the gait cycle, thus indicating some similarities of the temporal activation of both limbs.

The first module, which includes the knee extensors (*RF* and *VL*) and the hip extensor (*GluM*), was always activated at the beginning of the stance phase, first to absorb the negative work due to landing (weight acceptance) and then to start rising the centre of mass (Fig. 5). This is in accordance with previous results on walking (*Neptune, Clark & Kautz, 2009*; *Clark et al., 2010*; *Mehrabi, Schwartz & Steele, 2019*), and running (*Santuz et al., 2018b*; *Oliveira et al., 2016*).

In unilateral skipping, almost all the impact is absorbed by the trailing leg. Again, this agrees with the  high extensor moments at the hip joint in trailing legs during flight time and foot contact that have been reported (*Walter & Carrier, 2007*); this may explain why the activation of the weight acceptance in the trailing leg during unilateral skipping was significantly anticipated with respect to the leading leg and to the other gaits (synergy 1 in trailing leg, Fig. 5). In this sense, we suggest that the effort invested by synergy 1 in the trailing leg to reduce impact in landing could be one of the keys of the high metabolic cost of transport reported by other authors (*Minetti, 1998*; *Pavei, Biancardi & Minetti, 2015*).

Our results confirm the temporal pattern described in previous papers for symmetrical gaits (*Cappellini et al., 2006* ; *Saito et al., 2018a*; *Saito et al., 2018b*; *Neptune, Clark & Kautz, 2009*; *Santuz et al., 2018b*; *Oliveira et al., 2016*; *Clark et al., 2010*; *Mehrabi, Schwartz &*

*Steele, 2019*). According to *Cappellini et al. (2006)*, the difference between running and walking was the time shift of the activation component of the second synergy, associated to the plantar flexor muscles. The pattern of activation of this module in the leading limb during unilateral skipping was analogous to running (Fig. 5). Therefore, we may consider this behavior of the plantar flexor module as a distinctive characteristic between bouncing *versus* vaulting gaits. The anticipated activation was partially due to the shorter stance phase of bouncing gait with respect to walking: the calf muscles would provide forward propulsion and contribute to body support (*Neptune, Clark & Kautz, 2009*). *Ivanenko et al. (2008)* identified two distinct activations of the same set of muscles within a bilateral skipping cycle; they analysed EMG recorded from the right leg during bilateral skipping, where the same limb acts alternatively as trailing and leading, and this could explain the two peaks in an averaged cycle.

The ankle dorsiflexor tibialis anterior was included in the third synergy. Its *CoA* extended during the swing phase of walking (*Neptune, Clark & Kautz, 2009*) and running (*Oliveira et al., 2016*). In unilateral skipping, this module was anticipated to the very early swing phase, just after the toe-off. The fourth synergy was associated with the thigh extension, during the late swing and just before landing (Walking: (*Neptune, Clark & Kautz, 2009*; *Clark et al., 2010*); Running: (*Oliveira et al., 2016*; *Santuz et al., 2018b*). The activation timing of this synergy was shared among gaits. According to our second hypothesis, during unilateral skipping we found significant differences in *CoA* between the skipping trailing and leading legs in synergy 1 (weight acceptance) and synergy 2 (propulsion) (Fig. 5). The behavior of the two limbs is clearly different during the stance phase, but shares the same synergy pattern on the swing phase (C3 and C4, Fig. 5). The results in Fig. 5 refute our third hypothesis by showing very different activation patterns in the trailing leg than in walking, although the behavior of the leading leg in unilateral skipping is very similar to running.

## Center of mass trajectory

The effect of the synergic muscle work on the *COM* trayectory during weight acceptance and propulsion is shown by the 3D trajectory of the *COM* in Fig. 6 (*Minetti, Cisotti & Mian, 2011*). In walking, the propulsive phase accompanies the rise of the *COM* up to its highest point, while the weight acceptance begins with the heel strike and continues as the body switches from double to single support (Fig. 6A). In running, the propulsive phase ends with the take-off, while the *COM* continues its rise during the ballistic phase (Fig. 6B).

In unilateral skipping, the already mentioned asymmetry is evident (Fig. 6C). The anticipated weight acceptance phase prepares the trailing leg to support the impact after the flight phase. The two propulsion phases are very different, as in one case is a quasi-pendular weight transfer from trailing to leading while pushing forward, while the leading propulsion must push the up for takeoff.

## CONCLUSIONS

According to our hypothesis (i) and (ii), muscle synergies are shared across the repertoire of human locomotion and activation timing of trailing and leading limbs were different.

Therefore, in healthy people, unilateral skipping is not determined by different synergistic structure, but by adjustments in the basic motor patterns used in symmetrical gaits. The opportunity that the muscles have to activate within a stride cycle is what determines the biomechanical locomotor pattern of unilateral skipping, and not the modular structure. Our results, which at least partially refute our third hypothesis, support the idea of a mechanical paradigm different from those of walking and running. After having analyzed and comparing electromyographic data we can conclude that all human gaits present underlying features of muscle activation that suggest common neural control of locomotion.

## ACKNOWLEDGEMENTS

The authors acknowledge Dr. Renata Bona and Dr. Artur Bonezi for helping during the data collection. The authors also thanks Martín Arévalo for IT support; A. Saito and an anonymous referee for their critics and comments, which help to improve the paper.

### Funding

The authors received no funding for this work.

### Competing Interests

The authors declare there are no competing interests.

### Author Contributions

- Germán Pequera and Carlo M. Biancardi conceived and designed the experiments, performed the experiments, analyzed the data, prepared figures and/or tables, authored or reviewed drafts of the paper, and approved the final draft.
- Ignacio Ramírez Paulino analyzed the data, authored or reviewed drafts of the paper, and approved the final draft.

### Human Ethics

The following information was supplied relating to ethical approvals (i.e., approving body and any reference numbers):

The Ethics Committee of the Centro Universitario Regional Litoral Norte, Universidad de la República, granted Ethical approval to carry out the study (Exp. Nº311170-000521-18).

### Data Availability

The data is available in Zenodo: Pequera, G, & Biancardi, C. M. (2020). Human locomotion dataset [Data set]. Zenodo. Available at http://doi.org/10.5281/zenodo.4134977.

### Supplemental Information

Supplemental information for this article can be found online at http://dx.doi.org/10.7717/peerj.11970#supplemental-information.

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
