# Peer review of "Common motor patterns of asymmetrical and symmetrical bipedal gaits"

_PeerJ, doi:10.7717/peerj.11970_

## Round 0.1 · original submission · Major Revisions

Two reviewers found the paper ambitious but it has substantial perceived flaws in the statistical design, hypothesis testing and other aspects of the analysis in addition to structure. Please address these fully in a point-by-point Rebuttal with your revised manuscript. Thank you.

Reviewer 1 ·

Basic reporting

General Comments: This paper is well-written only at the former part of the whole manuscript, and it is interesting for us that whether the muscle control mechanisms are equivalent between the trailing and leading legs in different gaits. But some statements were not supported by your own data. There are many critical issues on this manuscript, in particular, statistics, results and the latter part of the discussion, and conclusion sections. The most significant issue is that this research completely lacks metabolic data, so I think that the structure of the 'story' should be re-constructed.

Specific comments;
1. In the conclusion section of the abstract, the authors stated about metabolic cost of skipping, but they did not measure metabolic data. This statement should be deleted.
2. Introduction is well-organized. Its research necessity and hypothesis are logically explained, but the discussion section does not specify what data is the most significant in your study.
3. In the methodological section, the authors measured EMG from 7 muscles. In particular, GM has two heads, so the authors should specify which head they focused.
4. Are these 7 muscles for EMG analysis are good enough compared to other previous studies?
5. Gait speeds were expressed as km/h. It should be replaced as m·sec-1.
6. There is a serious problem in statistics. You need to compare the data with two-way ANOVA. Why did the author use one-way ANOVA for comparing these data sets? I cannot agree with this statistics.
7. In line 307, a term “module 2 is module 3 in Fig.4. Please check this.
8. In this analysis, the authors used kinematic analysis. They quantified the external work (Wext), but internal work was not analyzed. Why not? I think that not only the inverted pendulum but also the recovery rate of the mechanical work may be able to support your idea why the energy cost of skipping is so expensive than other gait.
9. In Fig. 1, are there any significant differences? If there is, please put * or #.
10. F values are completely lacking from the result section.
11. I do not understand the linkage of each figure, because only Figs. 2, 4, and 7 appeared in the discussion section. I assumed that several figures should be in the supporting information.
12. Were your hypotheses established? In Line 370 and 443, you stated that 'According to…'. I assumed that both hypotheses were established. But what results (or Figure) support your statement?
13. The third hypothesis appeared in the conclusion section. The structure is highly confused.
14. I also feel strange at the conclusion section, because several articles are referenced. This section should be concluded based on your own data.
15. The latter part of your conclusion section never proved high energy cost of skipping. These statements should be deleted.
16. I could not find out any advanced interpretation based on your own data sets in the discussion section. The motor control study always requires neuromechanical evidence or physiological mechanisms to prove your hypothesis. Again, the research focus is good, but the data do not always support statements in the discussion and conclusion sections. For example, which synergies determine the metabolic cost of walking, running, and skipping? (although the metabolic data is lacking…) Are those equivalent among different gait even if the gait speed is the same at 6.5km/h? And why? Yes, synergies are necessary to simplify complicated muscle control systems during different gait patterns. So I think that the 'story' should be simplified whether the synergies happen at the same muscle regions during the same gait cycle among walking, running, and skipping.

Minor comments;
1. In line 408, ')' was doubly used. Check the spelling. This kind of minor mistake can be seen in Line 429 (need a space between ';' and 'they').
2. New lines are not necessary since Line 421 to Line 446. These are minor issues, but more elaborations should be conducted before submitting a manuscript.

Experimental design

Well structured, but the metabolic information is completely lacking.
As stated in the basic reporting, I highly concerned about statistics.

Validity of the findings

Viewpoint is interesting, but an advanced interpretation or newly-explained physiological mechanism is not found.

Additional comments

General Comments: This paper is well-written only at the former part of the whole manuscript, and it is interesting for us that whether the muscle control mechanisms are equivalent between the trailing and leading legs in different gaits. But some statements were not supported by your own data. There are many critical issues in this manuscript, in particular, statistics, results, and the latter part of the discussion, and conclusion sections. The most significant issue is that this research completely lacks metabolic data, so I think that the structure of the 'story' should be re-constructed.

Specific comments;
1. In the conclusion section of the abstract, the authors stated about the metabolic cost of skipping, but they did not measure metabolic data. This statement should be deleted.
2. Introduction is well-organized. Its research necessity and hypothesis are logically explained, but the discussion section does not specify what data is the most significant in your study.
3. In the methodological section, the authors measured EMG from 7 muscles. In particular, GM has two heads, so the authors should specify which head they focused on.
4. Are these 7 muscles for EMG analysis are good enough compared to other previous studies?
5. Gait speeds were expressed as km/h. It should be replaced as m·sec-1.
6. There is a serious problem in statistics. You need to compare the data with two-way ANOVA. Why did the author use one-way ANOVA for comparing these data sets? I cannot agree with the statistics used in this manuscript.
7. In line 307, a phrase "module 2" may be "module 3" in Fig.4. Please check this again.
8. In this analysis, the authors used kinematic analysis. They quantified the external work, but the internal work was not analyzed. Why not? I think that not only the inverted pendulum but also the recovery rate of the mechanical work may be able to support your idea why the energy cost of skipping is so expensive than other gaits.
9. In Fig. 1, are there any significant differences? If there is, please put * or #.
10. F values are completely lacking in the result section.
11. I do not understand the linkage of each figure. For example, only Figs. 2, 4, and 7 appeared in the discussion section. I assumed that several figures should be in the supporting information.
12. Were your hypotheses established? In Line 370 and 443, you stated that 'According to…'. I assumed that both hypotheses were established. But what results (or Figure) support your statement?
13. The third hypothesis appeared in the conclusion section. The structure is highly confused for me. Note that PeerJ is a multidisciplinary journal across wider research areas.
14. I also feel strange about the conclusion section, because several articles are referenced. This section should be concluded based on your own data.
15. The latter part of your conclusion section never proved the higher energy cost of skipping than other gaits. These statements should be deleted.
16. I could not find out any advanced interpretation based on your own data sets in the discussion section. The motor control study always requires neuromechanical evidence or physiological mechanisms to prove your hypothesis. Again, the research focus is good, but the data do not always support statements in the discussion and conclusion sections. For example, which synergies determine the metabolic cost of walking, running, and skipping? (although the metabolic data is lacking…). Also, are those equivalent among different gait even if the gait speed is the same at 6.5km/h? And why? Synergies are necessary to simplify complicated muscle control systems during any gait pattern. So I think that the 'story' should be simplified whether the synergies happen at the same muscle regions during the same gait cycle among walking, running, and skipping.

Minor comments;
1. In line 408, ')' was doubly used. Check the spelling. This kind of minor mistake can be seen in Line 429 (need a space between ';' and 'they').
2. New lines are not necessary from Line 421 to Line 446. These are minor issues, but more elaborations should be conducted before submitting a manuscript.

·

Basic reporting

Terminology should be improved throughout the manuscript. If raw data (enveloped EMG signals) is involved in the manuscript, it is suitable to help readers understand the methodological proceedings.

Experimental design

Data recording from bilateral legs is questionable. Statistical analysis should be improved. The current version uses parametric test for small sample size data.

Validity of the findings

Interpretation of findings should be more clearly written.

Additional comments

The manuscript examined the motor modules during skipping. I think that the purposes of the manuscript are significant in neurophysiological and motor control studies. However, I have some concerns regarding terminology and methodology. I suggest that the authors improve ambiguity and the manuscript should be more clearly written.

Major comments
1. Introduction is ambiguous and written redundantly. I suggest that the authors should make introduction section more briefly. For example, second paragraph in introduction, description regarding the metabolic cost of skipping is not very important for this study, because this study did not collect the data of energy expenditure during locomotion.

2. I could not understand how the authors collected EMG data during skipping. The authors should explain more clearly. EMG recordings were done from bilateral legs, but walking, running, and skipping are bilateral symmetric. So, the bilateral EMG recordings are not need to compare the motor modules between locomotor movements. It is enough to collect from one leg. Please justify your study design about EMG recording. In addition, I was confused “preferred leg”, “leading leg”, and “trailing leg” during skipping. In the case of stepping over an obstacle, analyzing differences in muscle activation between leading and trailing legs are significant. However, the authors compared the motor modules between walking, running, leading leg, and trailing leg. The authors need to explain the EMG processing in this study how the EMG data in leading and trailing legs during skipping was sampled showing a typical example.

3. Statistical analysis is questionable. The number of the extracted modules by NNMF in locomotion is different between speeds, locomotor movements, and types of modules. As shown in figure 2, sample size (the number of synergies for each type of module) is not fixed. Thus, I suggest that non-parametric test is appreciate in this study. Furthermore, I could not understand why the authors done one sample t-test to sort the type of modules. Similarity of the modules across gaits and speeds was evaluated by cosine similarity. Is the maximal value of cosine similarity not enough to sort the modules?

Minor comments
Abstract
L49, Is skipping asymmetry movements?
L50, This sentence is method.
L52, How did you identify motor modules from EMG data? The authors should write it in method section.

Introduction
L80, Movement of skip is bilateral asymmetry?
L106, Mistake citation, Yokoyama et al., 2019.
L117, Mistake citation, Saito et al., 2018b is running study.

Methods
L167, Foot note is expressed as number.
L215, Which is the preferred limb? Preferred limb means dominant limb? How did you identify the preferred limb during walking and running?
L216, The preferred limb was used as trailing leg. What meaning of this sentence?

Results
L275, The authors show difference in gait parameters among different types of locomotion. Why don’t you compare the parameters between bilateral legs during walking or running? Were gait parameters similar between leading and trailing limbs? In the manuscript, the authors demonstrate walking and running are symmetry.
L294, The authors used 4 types of modules for analysis. How many modules are involved in each averaged data of Figure 3? The authors should show the number of modules (sample size) which included in mean of activation timing and weighting.

Discussion
L338, I was confused that much terms exist in the manuscript such as trailing leg, leading leg, stance phase, forelimb, hindlimb. What is different between these terms. I suggest that the authors should define these terms in introduction or method sections.
L360, This sentence is not necessary.
L386, The extracted synergies between present and previous study were shared, but the number of EMG recordings between present and previous works were quite different. Capellini et al. recorded 32 muscles included lower-limb and trunk muscles and Yokoyama et al. recorded 16 muscles during locomotion. This is methodological issue in this study. The authors should discuss effect of number of EMGs on the motor module extraction by NNMF.

---

## Round 0.2 · Major Revisions

The reviewers still find substantial issues with the presentation and key scientific aspects of the study. Renaming "skipping" as "“Unilateral Skipping” would be better than "galloping" as the latter is a quadrupedal gait, not bipedal. This would aid clarity. Showing a " typical example of raw data" as recommended appears necessary. A reviewer challenges that with N=4 you should remove all "mechanical and metabolic data" and just present the EMG data. This is a reasonable suggestion and the EMG data alone would still be publishable. PeerJ does not have word limits for papers, but you should carefully re-examine your wording for redundancy as recommended, and where it would help the reader, shorten the MS; removing mechanical and metabolic data would aid that. Ensure that all points are addressed in your Rebuttal and in the main MS where possible.

Reviewer 1 ·

Basic reporting

Many statements are still redundant. Also, if you want to refer to the mechanical efficiency, you needed to measure metabolic data from all 14 participants. As you may know, there is a big gap between muscular synergies and locomotion efficiency, even if you succeeded to take the metabolic data from all participants. You can and should re-construct the 'story' of the whole manuscript.

Experimental design

More ingenuity is necessary.

Validity of the findings

The story of this paper cluttered so much, so it is very difficult for me to find out where the most important part is. Please remind that you stated more than 6400 words in this paper. That is too many. At most, 4500-5000 words including references (e.g. Steele, Tresch, & Perreault, (2013) in the main text) are good enough to examine your study questions.

Additional comments

The story of this paper still cluttered so much, and it has not yet got together. Many statements are still redundant. My recommendations for your future submission are as follows.

1) Please focus only on the EMG data obtained from three different gaits. Mechanical and metabolic data are not necessary any more, because you could obtain only 4 metabolic data.
* Are mechanical and metabolic data truly necessary to examine your original research questions?

2) A simple comparison can be conducted only at 1.8 m/sec.
** Other gait speeds are not comparative among three different gaits, so that these data should be in the supplemental information. If so, anybody can understand 1-way ANOVA is an appropriate statistical method.

3) Many redundant statements should be deleted to complete within 4000-4500 words.
** For example, L121-L134 and L492-L503 are not related to your main focus.

4) Are all figures necessary in this paper?

·

Basic reporting

Terminology should be improved in the manuscript. Raw EMG data is needed to understand the methodology.

Experimental design

no comment

Validity of the findings

no comment

Additional comments

Manuscript was improved more clearly. However, current version of the manuscript is difficult to understand yet. I have some concerns to improve the manuscript below.
Specific comments
1. I could understand what skipping mean in the manuscript. However, “Skipping” will confuse the readers yet. I suggest that the authors use “Galloping” or “Unilateral Skipping” instead of “skipping” throughout the manuscript.

2. The authors should explain how individual participants familiarized galloping before the experiment. I expect that some participants did galloping awkwardly. Or, all participants could galloping well without any trials?

3. The authors should show typical example of raw data (i.e., envelop EMG signals) as a function of the percentage of a step cycle. I suggest that the figure involving the illustration of gallop-like-skipping movement is suitable to help readers understand the methodologies in this study.

---

## Round 0.3 · accepted · Accept

I have checked the revision as has one of the two prior reviewers; the most critical previous reviewer was unavailable. However, the two of us agree that the paper is satisfactory now and can be accepted-- congratulations indeed. Your attentive efforts paid off.

·

Basic reporting

no comment

Experimental design

no comment

Validity of the findings

no comment

Additional comments

no comment